# The Influence of Omeprazole on the Dissolution Processes of pH-Dependent Magnetic Tablets Assessed by Pharmacomagnetography

**DOI:** 10.3390/pharmaceutics13081274

**Published:** 2021-08-17

**Authors:** Guilherme A. Soares, Deivid W. Pires, Leonardo A. Pinto, Gustavo S. Rodrigues, André G. Prospero, Gabriel G. A. Biasotti, Gabriela N. Bittencourt, Erick G. Stoppa, Luciana A. Corá, Ricardo B. Oliveira, José R. A. Miranda

**Affiliations:** 1Department of Biophysics and Pharmacology, Institute of Biosciences, São Paulo State University—UNESP, Botucatu 18618-689, São Paulo, Brazil; deivid.pires@unesp.br (D.W.P.); leonardo.antonio@unesp.br (L.A.P.); gustavo.serafim@unesp.br (G.S.R.); andre.prospero@unesp.br (A.G.P.); gabriel_gdab@hotmail.com (G.G.A.B.); ga.nbittencourt@gmail.com (G.N.B.); e.stoppa@unesp.br (E.G.S.); jose.r.miranda@unesp.br (J.R.A.M.); 2Center of Integrative Sciences, Alagoas State University of Health Sciences—UNCISAL, Maceio 57010-382, Alagoas, Brazil; luciana.cora@uncisal.edu.br; 3School of Medicine of Ribeirão Preto, University of São Paulo, Ribeirão Preto 18618-689, São Paulo, Brazil; rbdolive@fmrp.usp.br

**Keywords:** AC biosusceptometry, Eudragit^®^ E-100, pharmacomagnetography, proton pump inhibitor

## Abstract

Pharmacomagnetography involves the simultaneous assessment of solid dosage forms (SDFs) in the human gastrointestinal (GI) tract and the drug plasmatic concentration, using a biomagnetic technique and pharmacokinetics analysis. This multi-instrumental approach helps the evaluation, as GI variables can interfere with the drug delivery processes. This study aimed to employ pharmacomagnetography to evaluate the influence of omeprazole on the drug release and absorption of metronidazole administered orally in magnetic-coated tablets. Magnetic-coated tablets, coated with Eudragit^®^ E-100 (E100) and containing 100 mg of metronidazole, were produced. For the in vivo experiments, 12 volunteers participated in the two phases of the study (placebo and omeprazole) on different days to assess the bioavailability of metronidazole. The results indicated a shift as the pH of the solution increased and a delay in the dissolution of metronidazole, showing that the pH increase interferes with the release processes of tablets coated with E100. Our study reinforced the advantages of pharmacomagnetography as a tool to perform a multi-instrumental correlation analysis of the disintegration process and the bioavailability of drugs.

## 1. Introduction

The oral route is the simplest, most convenient, patient compliant, and safest drug administration mode [1,2]. Generally, a drug administered orally is in a solid dosage form (SDF), such as gastric soluble tablets. SDFs have the primary objective of providing immediate drug release or promoting release in specific locations to be absorbed in the gastrointestinal (GI) tract [3].

However, drug delivery systems must overcome highly variable physiological and physicochemical conditions throughout the GI tract, including the luminal pH [4,5,6]. Several factors can modify the gastric environment’s pH, especially those related to drug co-administration, such as proton pump inhibitors (PPIs) [7,8,9]. PPIs (such as omeprazole) inhibit gastric acid secretion in the parietal cells of the stomach. Consequently, PPIs increase the intragastric pH > 4, and they can directly interfere with the absorption of drugs taken simultaneously with this type of medication. Studies show that co-administration of drugs with omeprazole modifies their pharmacokinetic parameters, such as absorption, bioavailability, distribution, and metabolism [10,11,12,13,14,15].

In addition to directly affecting absorption and bioavailability, the change in pH can also change the performance of pH-dependent coating strategies. A common strategy to achieve specificity of pH-dependent drug delivery systems is coating SDFs with polymers with a pH-dependent solubility, such as Eudragit^®^ E-100 (E100) [16,17,18]. E100 is a cationic polyelectrolyte based on dimethylaminoethyl methacrylate and other neutral methacrylic acid esters. Due to E100’s pH-dependence characteristics, this polymer can be used to release the drug in the gastric environment immediately and perform several other interesting applications.

Although in vitro tests performed during the development of formulations are quite effective, in vivo tests are essential to evaluate the behavior of pharmaceutical forms in the human GI tract. It is of paramount importance to implement methods capable of characterizing pharmaceutical forms in vivo. Biomagnetic methods are non-invasive and have been implemented as a viable and promising pharmaceutical research tool, allowing correlations between monitoring the pharmaceutical forms along the GI tract and pharmacokinetic parameters [19,20]. Alternate Current Biosusceptometry (ACB) is a biomagnetic technique employed as an alternative method in pharmaceutical research to evaluate the behavior of solid dosage forms in vitro, monitoring their journey in the different segments of the GI tract in vivo. It is also an auxiliary tool for SDFs quality control [18,20,21].

Recently, ACB was proposed in association with pharmacokinetics in an approach called pharmacomagnetography. Pharmacomagnetography consists of relating magnetic monitoring of the pharmaceutical form with the pharmacokinetic parameters and bioavailability of the drug. A previous study showed a successful approach using the ACB system to evaluate the GI transit and pharmacokinetics profile of magnetic enteric-coated tablets and assess the drug delivery and bioavailability [22].

In this context, this study aimed to use this approach to assess the influence of gastric pH changes caused by treatment with PPI on the process of releasing a model drug administered orally in tablets coated with E100.

## 2. Materials and Methods

### 2.1. Materials

All the materials used in this study were of analytical grade. Manganese ferrite (MnFe_2_O_4;_ diameter of 50 < ϕ < 75 µm) was purchased from Ferroxcube, El Paso, TX, USA. Metronidazole, isopropyl alcohol (IPA), acetonitrile, acetone, polyethylene glycol 6000, magnesium stearate, and talc were purchased from Sigma-Aldrich, Santo Andre, Brazil. Microcrystalline cellulose (Microcel^®^) and sodium starch glycolate (Explosol^®^) were kindly supplied by Blanver, Taboão da Serra, Brazil. Colloidal silica (Aerosil^®^) was purchased from Evonik, São Paulo, Brazil. E100 (Degussa, Rohm Pharma Polymers) was provided by Almapal Tecnologia Validada, Cotia, Brazil.

### 2.2. pH-Dependent Magnetic Coated Tablets

pH-dependent magnetic coated tablets were manufactured to be used in this study, since this formulation was not intended for commercial purposes. The magnetic tablets were obtained by direct compression in a single-punch die set machine (Marconi MA098/C, Piracicaba, Brazil), using a 12 mm diameter matrix. The tablets were prepared using 500 mg of manganese ferrite microparticles, 100 mg of metronidazole as a model drug, 300 mg of microcrystalline cellulose, 10 mg of magnesium stearate, 50 mg of Aerosil^®^, and 40 mg of Explosol^®^. All the excipients and the drug were mixed and manually compressed at a pressure of 30 kN. The tablets were coated by the spray-drying method in a bench-coating machine (LEMAQ, São Paulo, Brazil), with a solution containing Eudragit^®^ E-100 as a pH-dependent coating, maintaining the stability of the tablet and the release of the drug immediately in gastric media. The coating solution was prepared as follows: E100 (50%, *w*/*w*) was dispersed in isopropyl alcohol (25%, *w*/*w*) and acetone (25%, *w*/*w*). This solution was mixed until all of the polymers were dissolved. Polyethylene glycol 6000 33% (12.2%, *w*/*w*), as a plasticizer, and talc (37.8%, *w*/*w*), as a lubricant, were dispersed in isopropyl alcohol (25%, *w*/*w*) and acetone (25%, *w*/*w*) and mixed for 10 min. After this procedure, the polyethylene glycol 33% and talc solution were added to the solution containing the E100 and mixed until it became a homogeneous solution. In the end, this solution was filtered through a 0.5 mm filter and subsequently applied to the tablets to obtain a 6% weight gain. The coating process was carried out at an inlet temperature of 35 ± 5 °C, while the outlet temperature was 30 ± 4 °C. The spray rate of the coating solution was 15 mL/min, and the atomization pressure was 1.5 bar. When the tablets reached a gain of 6% (*w*/*w*) of the initial weight, they were placed in an oven at a temperature of 40 °C for 24 h for drying and maturation of the applied polymer. In the present study, magnetic-coated tablet formulations were tested according to the outline in the US Pharmacopeia. Therefore, the average weight, hardness, and friability were assessed for all tablet formulations. The disintegration and dissolution tests are provided in the manuscript.

The average weight was calculated by weighing the tablet batches individually. We used a 10^−4^ g resolution analytical balance. This test was carried out to check the uniformity of the tablets in each formulation, in which the acceptable criteria variation limit was ±5%.

The hardness test was performed using a durometer Dr. Schleuniger model 6D (Pharmatron, Westborough, Ma, EUA). The test results were obtained and represented by six assessments for each tablet, in agreement with U.S. Pharmacopeia (USP), which sets the minimum acceptable hardness in the 4 to 10 kilopounds (Kp) range.

Regarding friability, the test allowed the determination of the resistance of the tablets to abrasion when subjected to the mechanical action of specific equipment. The friability of the tablets was determined using an automated system, Friabilometer EF-2 (Nova Ética, São Paulo, Brazil), at 25 rpm for four minutes. After this, they were weighed, and the weight loss (%) was calculated according to the USP specifications, in which the friability must not exceed 1.5% of tablet weight. The percentage of weight loss was calculated with the Equation (1):(1)$$\mathrm{Friability}\;\left(\%\right)=\frac{\mathrm{initial}\;\mathrm{weight}\;-\mathrm{final}\;\mathrm{weight}\;}{\mathrm{initial}\;\mathrm{weight}}\times 100$$

### 2.3. Alternate Current Biosusceptometry

AC Biosusceptometry is a biomagnetic technique based on using induction coils to magnetize and detect the magnetization of magnetic materials. This study employed two ACB setups: single and multichannel systems. The single-channel ACB (SC-ACB) has one pair of induction and detection coils (ϕ = 2.9 cm), and the multichannel ACB (MC-ACB) system has one pair of induction coils (ϕ = 11 cm) and seven pairs of detection coils (ϕ = 3.5 cm). The induction coils of the SC-ACB are supplied with 500 mA and the MC-ACB with 150 mA, both with AC voltage at 10 kHz, and the signal is acquired at a rate of 20 Hz in a LabView environment. The ACB system presents a high temporal resolution, which depends only on the acquisition frequency and the A/D board capability. Here, the temporal resolution obtained by both ACB setups was 50 ms.

Regarding the spatial resolution, the obtained SC-ACB and MC-ACB spatial resolutions in these experiments were in the order of millimeters and centimeters, respectively. In in vivo situations, in which the ACB detection surface was positioned at around 4 cm until the gastric projection, the spatial resolution decreased, and the SC-ACB and MC-ACB were 2 cm and 4 cm, respectively. The SC-ACB had a sensitivity limit of about mg per cm^3^ and the MC-ACB of about g per cm^3^.

Due to its versatility, the SC-ACB was employed to evaluate the performance of the tablets’ transit and determine the value of the Gastric Retention Time (GRT, the time interval between ingestion and elimination of the stomach) of the coated magnetic tablets by manually scanning a given region of interest [21]. MC-ACB is suitable for simultaneous signal acquisition in seven different points of space. This modality was used to obtain magnetic images in real-time [20,23] and evaluate the disintegration process of the coated magnetic tablets. Figure 1 presents the two ACB systems and their operation principles.

### 2.4. In Vitro Studies

#### 2.4.1. USP 2 Dissolution Procedure

The dissolution studies of the coated magnetic tablets were carried out in a USP dissolution test apparatus 2 (paddle method, 50 rpm, 37 ± 0.5 °C). A simulated gastric fluid with different pH values was used as a dissolution medium. The media volume was 900 mL. An exact sample volume (1 mL) was withdrawn and replaced with a fresh dissolution medium at times: 0.5, 1, 2, 3, 4, 5, 10, 15, 20, 25, 30, 45, 60, 75, 90, 105, and 120 min. The time intervals were projected to evaluate the initial metronidazole release at all pH values. All samples were filtered through a cellulose acetate membrane of 0.22 µm.

All tests were performed in sextuplicate, and samples were analyzed with a UV-spectrophotometer-Ultrospec 2000Spectrophotometer (Pharmacia Biotech, Uppsala, Sweden) at 277 nm (pH 1.2 and pH 2.0) or 318 nm (pH 3.0 and pH 4.5). 

Metronidazole concentrations were calculated using calibration profiles based on absorbance versus concentration curves, which were previously standardized. The lag time $T_{lag}$ (in min) was calculated as the time point at which metronidazole was first detected in the dissolution medium.

The dissolution efficiency (DE) was calculated as the percentage ratio between the area under the dissolution curve up to time t ($AUC_{\;0-t}$) and the area of the rectangle described by 100% dissolution at the same time point ($A_{t}$), and it was defined as follows [24] in Equation (2):(2)$$DE=\frac{AUC_{0-t}}{A_{t}}$$

We established a validation protocol for the metronidazole characterization and quantification by UV-spectrophotometer. This validation was based on assessing a series of validation parameters, such as the Limit of detection (LOD), Limit of quantification (LOQ), sensitivity, accuracy, and linearity. For the UV-spectrophotometer and our metronidazole reference used, we found a LOD of 0.070453 µg/µL and a LOQ of 0.234843 µg/µL. The results demonstrated the development and validation for our UV-spectrophotometer quantification of metronidazole under the International Harmonization Conference (ICH, 2005) and the National Agency of Sanitary Vigilance (ANVISA, 2003) guidelines. According to the results, the method is linear, specific, and it can accurately measure the drug in the selected range with low uncertainty.

#### 2.4.2. Multichannel ACB System Measurements (MC-ACB)

Using a squared glass vessel with the same dissolution mediums as used for the *USP 2 dissolution procedure,* we designed an experiment to investigate the suitability of the MC-ACB system, to assess the pH dependence of the tablets’ E100 coat dissolution through magnetic signal reconstructed images. The glass vessel was positioned in front of the MC-ACB system at an axial distance of 3 cm, and then, the coated magnetic tablets were introduced into the solution, and signals were acquired for 30 min. This phase was performed in sextuplicate. As demonstrated in our previous studies, the area in the magnetic images obtained from the signals was evaluated to determine the disintegration process [18,25]. The magnetic image reconstruction and analysis are further described in the ‘magnetic data analysis’ section.

### 2.5. In Vivo Study Protocol

Twelve healthy volunteers (six males, six females, aged between 20 and 27 years, body weight between 50 and 80 kg, and BMI < 22 kg/m²) were enrolled in this study. General exclusion criteria included pregnancy, smoking, abdominal surgery, diabetes, other chronic endocrine disorders affecting GI motility, and treatment with prokinetics, anticholinergics, opiates, or macrolide drugs. Participants remained moderately active during the study day and received standardized meals at 2 h, 4 h, and 6 h post-dose [22].

This was a single-center, randomized, double-blind, comparative study to evaluate the pharmacokinetic and magnetic behavior of coated magnetic tablets containing metronidazole before (placebo) and after omeprazole administration. There was a 7-day period separating both treatments for the same subject. Before the experiment, the volunteers received either 40 mg of placebo or 40 mg of immediate-release omeprazole at night. After fasting overnight, 30 minutes before the beginning of the experiment, all volunteers received another dosage of the same tablet as previously administered. The subjects swallowed the E100-coated magnetic tablets with 200 mL of water, and the magnetic signals were recorded for 30 min through the MC-ACB. The volunteers were standing, and the system was positioned on the volunteers’ abdomens, in the region corresponding to the gastric projection according to external anatomical references [25]. After continuous recording with the MC-ACB system, mapping from the abdominal surface was performed every 30 min using the SC-ACB system to locate the coated magnetic tablet and determine the GRT.

Blood samples were obtained at 30 (pre-dose), 3, 6, 9, 12, 15, 18, 21, 24, 27, 30, 45, 60, 90, 120, 150, 180, 210, 240, and 300 min after tablet administration to assess the absorption profile of metronidazole. The samples were immediately centrifuged, and the serum was stored at −80 °C until analysis.

This study was approved according to the protocol approved by the Ethics in Research of the Medical School, State University of São Paulo (UNESP). It was conducted under the Declaration of Helsinki and its revisions. All volunteers gave written informed consent before the study. (Certificate of Presentation for Ethical Appreciation: 41563015.3.0000.5411, Committee’s technical opinion number: 986486, Data of Approval: 4 May 2015).

### 2.6. Pharmacokinetic Analysis

The metronidazole quantification in serum samples was performed using ultra-high-performance liquid chromatography (UHPLC) through an Acquity H Class system equipped with a UV-vis (λ = 277 nm) Acquity UPLC Photodiode Array (Waters^®^, Milford, CT, USA). The metronidazole extraction was performed with a previously established protocol [26]. Briefly, 2 mL of acetonitrile was added to 500 µL of serum and mixed for 2 min. Samples were then centrifuged at 4500 rpm at 4 °C for 10 min, and the supernatant was transferred to a 15 mL tube, subsequently frozen at −80 °C, and lyophilized overnight. The lyophilized samples were reconstituted on the following day in 200 µL of a solution of water/methanol (90:10, *v*/*v*), and then centrifuged for 10 min at 4 °C and 14,000 rpm. The sample measurements were conducted in the reverse-phase using an Acquity UPLC^®^ BEH C18 column (1.7 µm, 2.1 × 50 mm, Waters^®^, Milford, CT, USA), with an automatic injector, maintained at 25 ± 1 °C. The mobile phase was eluted in an isocratic way in a flow of 0.5 mL/min and composed of water/acetonitrile (90:10, *v*/*v*).

The following pharmacokinetic parameters were obtained: $T_{lag}$, defined as the time at which the first detection of metronidazole in plasma occured; $C_{max}$, defined as the maximum plasma concentration of metronidazole; $T_{max}$, defined as the time taken to reach $C_{max}$; and $AUC_{0-300}$, defined as the area under the plasma concentration curve versus time, from time t=0 to t=300 min.

### 2.7. Magnetic Data Analysis

The magnetic signals recorded by the MC-ACB system were obtained as a time series matrix, and magnetic images were reconstructed by fitting the data of each sensor to the system transfer function. Image processing for quantification included the spline method interpolation, background subtraction, brightness and contrast adjustment, and segmentation. The segmentation quantified the number of pixels in the determined region of interest (ROI) [25].

The coated magnetic tablets’ in vitro and in vivo disintegration processes were characterized by a magnetic marker (MM) transition to a magnetic tracer (MT). In the magnetic images, the MM is represented as a delineated point, while the spreading of the magnetic material characterizes the MT, thus increasing the imaging area. The onset of the disintegration process was calculated as the meantime disintegration ($T_{50}$), which represented a 50% increase of pixels in the imaging area [27]. In the case that the imaging area variation was less than 50% during the 30 min of magnetic signal acquisition, $T_{50}$ was defined as not available, since a magnetic measurement was performed within this 30-min interval. It is worth mentioning that the interval time of $T_{50}$ was determined according to the protocol adopted using MC-ACB [21,28].

The SC-ACB mapping was performed to verify if the tablets were in the stomach, through signal intensity to quantify the GRT [21]. An SC-ACB scan could also be used to assess the behavior of magnetic coating tablets to determine gastric emptying.

### 2.8. Statistical Analysis

The magnetic signals were analyzed in MatLab (Mathworks, Natick, MA, USA) and Origin (Version 2016, OriginLab Corporation, Northampton, MA, USA), and the data are presented as the mean ± standard deviation. The disintegration process and drug dissolution test of the coated magnetic tablets obtained in vitro at different pH mediums were compared by one-way analysis of variance (ANOVA) followed by Tukey’s multiple comparison test. In vivo, magnetic and pharmacokinetic parameters between both treatments were compared using the paired Student *t*-test. Values of *p* < 0.05 were considered to be statistically significant. All data analysis and statistics were performed with GraphPad Prism (GraphPad Software, La Jolla, CA, USA).

## 3. Results

### 3.1. Tablet Quality Control Assays

The tablet quality control assays recommended by USP allowed the evaluation of the physical characteristics regarding the uniformity and properties of mechanical resistance, besides the average weight of non-coated tablets. The average mass (1.004 ± 0.003 g), the average diameter (12.12 ± 0.01 mm), and average height (6.08 ± 0.04 mm) adhered to the maximum variation of 5% allowed. The friability and hardness test results were 0.18% and 6.43 ± 0.74, respectively. Both results met the USP recommendations.

### 3.2. In Vitro Studies

A representative plot of the metronidazole release percentage in time for different values of pH dissolution media is shown in Figure 2. Even though 100 mg of metronidazole release was observed for all pH mediums during the evaluation, a positive shift (in time) was observed as the pH increased. Based on individual dissolution profiles, $T_{lag}$ and DE were calculated, and the results are expressed in Table 1. The pH 3.0 dissolution media resulted in an increased $T_{lag}$ in comparison to pH 1.2, whereas the pH 4.5 dissolution medium delayed the $T_{lag}$ in comparison to all other pH dissolution media. The DE values showed no statistically significant difference among the different pH dissolution media. These results were expected due to the pH dependence of the E100 coating.

A representative plot of the imaging area (pixels) versus time for distinct pH mediums obtained by MC-ACB images is shown in Figure 3. As observed in Figure 1, an increase in the pH value resulted in a positive shift in the time for image area growth. Table 2 presents the quantification of onset disintegration ($T_{50}$) for each pH. The pH 2.0 medium delayed the onset of disintegration in comparison to pH 1.2. The pH 3.0 medium resulted in a delayed $T_{50}$ in comparison to pH 1.2 and 2.0, whereas $t_{50}$ values for pH 4.5 were delayed in comparison to pH 1.2, 2.0, and 3.0. All coated magnetic tablets were disintegrated entirely during the measurement time, regardless of the pH value. The $T_{lag}$ and $T_{50}$ values obtained in the USP 2 dissolution procedure and MC-ACB images showed a Pearson’s correlation coefficient of 0.83, representing good linearity between the methods [29]. With this, the MC-ACB imaging method showed effectiveness in assessing the influence of medium pH in the coat dissolution in vitro process and suggests the suitability of using magnetic images to analyze the disintegration process for in vivo measurements.

In vitro studies may be considered an alternative to in vivo pharmacokinetic studies. In vitro dissolution tests can be an essential tool for the pharmaceutical development of solid oral dosage forms, either generic or new drugs. Depending on the characteristics of the intended product and the performance in vitro, the results may be required for waiving the regulatory requirement for in vivo Bioavailability (BA) and Bioequivalence (BE) studies [30]. Although in vivo studies are vital in drug development, the predictive dissolution methodology eliminates unnecessary steps involving volunteers’ exposure in clinical studies, reducing financial expenses [31].

The ACB system’s assessment through the acquisition of magnetic signals of tablets is a valuable methodology to characterize the tablets’ performance in vitro and in vivo in the GI tract. Thus, this approach is interesting for in vitro investigations, and it could consolidate future studies aiming to obtain biowaivers for new and generic drugs. It is worth noting that the ACB system is an alternative tool, like other methodologies, that can carry out a non-invasive analysis to assess the performances of drug formulations. Although the ACB does not provide insights into the drugs’ molecular state and chemical stability, the association with spectroscopy methods, such as Raman spectroscopy and near-infrared spectroscopy (NIR), may be very interesting.

The methodologies have been applied to the advanced characterization of drug delivery systems, allowing quality assessment of tablets by non-destructive assay.

In addition, nuclear magnetic resonance (NMR) has presented itself as a suitable technique for obtaining information for developing controlled-release systems due to its nondestructive and noninvasive aspects.

### 3.3. In Vivo study Protocols

Table 3 shows the pharmacokinetics and quantified parameters of the magnetic images of each volunteer. After administering the placebo, the E100 coating of the magnetic tablets was successfully dissolved in all volunteers during the MC-ACB measurement (30 min); the expected fasting gastric pH was approximately pH 2.0 [32]. In contrast, the results obtained after omeprazole administration showed that in 75% of volunteers (2, 3, 4, 5, 6, 8, 9, 10, and 11), the coated magnetic tablets remained intact for 30 min in gastric fluid, as a result of the increased pH. In 56% of cases (volunteers 2, 3, 5, 8, and 11), no metronidazole was detected in their plasma samples during the 300 (5 h) min of measurements. In the other 44% of cases (volunteers 4, 6, 9, and 10), metronidazole was detected in their plasma samples after the end of the MC-ACB signal acquisition (i.e., 30 min after administration). For these cases, the $T_{50}$ was considered as not available. Although the results indicated an increasing trend in several volunteers, the mean of the $T_{lag}$ was not significantly different between the control and omeprazole groups, mainly due to the variation between individuals. For example, there was no release of metronidazole in five volunteers, which prevented a comparative group analysis. Furthermore, the individual values in Table 3 show that $T_{max}\;and$ $C_{max}$ did not present differences. Regarding $C_{max}$, the results were in agreement with the fact that metronidazole does not undergo inactivation under pH variations [33]. On the other hand, the administration of omeprazole induced a significantly lower metronidazole bioavailability response, which was significantly lower at intervals of $AUC_{0-60min}$ (*p* = 0.001) and $AUC_{0-300min}$ (*p* = 0.05). As the AUC indicated significant differences in the extent to which the drug becomes available in the bloodstream, the effect of omeprazole once co-administered with E100-coated tablets was evident in most subjects.

The assessment of GRT values by SC-ACB mapping of the tablets, combined with the pharmacokinetic parameters, showed that pharmacomagnetography analysis is crucial for obtaining a complete understanding of the drug release process, and it cannot be assessed with a one method evaluation.

For example, in volunteer 6, the coated magnetic tablet remained intact in the stomach for 120 min (GRT data), and the $T_{lag}$ obtained was 150 min in the omeprazole treatment. In this case, we can be sure that the coated tablet did not dissolve in the stomach, based on the magnetic data collected. Therefore, pharmacomagnetography analysis may state that the coating was dissolved or that the tablet was not in the stomach.

In the omeprazole treatment of volunteer 6, the metronidazole was probably released in the small intestine due to peristaltic movements inducing fissures in the tablet’s coating. As metronidazole has high solubility and intestinal permeability, the absorption could have occurred at the same release site, which a previous study endorsed.

Table 3 shows that inter-individual biological variability data interfered with our analysis. We obtained profiles in which the administration of omeprazole did not provide any effect on pH increase. In volunteer 7, for example, $T_{\mathrm{lag}}$ and $T_{50}$ indicated values in line with the Control Phase parameters, likely due to the non-increased pH following omeprazole. On the other hand, we observed cases in which the magnetic tablet was directly emptied, neither released in the stomach, nor absorbed at any site of the TGI. The parameters of volunteers 2 and 3 suggest these situations; most of the parameters $T_{\mathrm{lag}}T_{\max},{\;C}_{\max},{\;\mathrm{AUC}}_{0-60}$, and ${\mathrm{AUC}}_{0-300\;}$ presented no value. Thus, we concluded from the GRT data (40 min) that the magnetic tablet was quickly emptied. [34]. Katz et al. (2001) [35] reported a large inter-subject variation of intragastric pH after administration of omeprazole 20 mg, which is the crucial factor of pH-dependent coating performance. Even though no intragastric pH measurements were performed in this work, the data strongly suggest that, in the cases in which no metronidazole was found in the plasma after administration of omeprazole, the intragastric pH was likely above 5, which is the upper limit for E100 solubility. 

According to the Biopharmaceutical Classification System (BCS), metronidazole is a class I drug, presenting high solubility and high intestinal permeability, which may show in plasma after some time, even with incorrect release sites. The pharmacokinetics of metronidazole are also stable after intravenous or oral administration of omeprazole; even if the pH of the dissolution medium increases, the drug’s solubility rate does not decrease [33,36]. Future studies may address the effects of omeprazole administration concomitantly with BCS class II or IV drugs, which will present a low intestinal permeability. Even considering that many drugs have low permeability regardless of PPIs, it is important to properly evaluate the potential interaction and clinical relevance of co-administration with gastric pH-elevating agents since it is common for cancer patients to use these agents, for example. Therefore, studies aiming to assess anticancer availability impacts are essential for determining less potent PPIs or medical strategies to use concomitant PPIs with any drug [37,38]. Pharmacomagnetography analysis in these cases (BCS class II and IV drugs) would be interesting in order to assess the location of drug absorption. Regarding BCS class II drugs, there have been broad investigations of the Hedgehog (Hh) pathway [39,40], which have yielded US FDA approval of a few Hh inhibitors (Hh-I) as promising alternative treatments for basal cell carcinomas [41]. Sonidegib, an example of a Hh-I, is classified as a BCS class II drug and has pH-dependent aqueous solubility, with lower solubility at higher pH levels. Several studies have been performed to assess the influence of PPIs on this inhibitor. In some of these, when coadministered with esomeprazole, the (S)-isomer of omeprazole, the AUC and the $C_{max}$ of Sonidegib was decreased by 32 to 38%, respectively [42]. Another study highlighted the challenge for scientists aiming to offer new cancer therapies that are not susceptible to acid-reducing agents (ARAs), including PPIs [43].

Infection of the stomach caused by *Helicobacter pylori* infections is responsible for most peptic ulcers and chronic gastritis and is usually treated with triple therapies, which consist of omeprazole and amoxicillin with either clarithromycin or metronidazole, if the patient is allergic to penicillin and derivates [44]. Both clarithromycin and metronidazole have a bitter taste [45,46]; therefore, coating formulations, such as E100, may be used for taste masking [47,48,49]. The results presented here highlight the importance of avoiding a pH-dependent coating in cases when omeprazole is administered concomitantly. Moreover, PPIs are widely prescribed for all gastric pH control cases, even when their benefit is considered low or even potentially harmful with adverse outcomes, particularly for patients taking an antiviral. Figure 4 shows examples of the magnetic images and pharmacokinetic analysis of the coated magnetic tablet administered (volunteer 3) after the placebo. Immediately after the administration (Figure 4A, t = 0 min), the tablet was observed as a well-defined circular region point, since the coating was initially intact. At the end of the measurement (Figure 4B, t = 30 min), the coating was successfully dissolved, which was shown by the spreading of the magnetic material increasing the image area. The area of the magnetic image and serum metronidazole concentration are shown in Figure 4E. Since the drug is released after coat dissolution, the increase of the metronidazole plasmatic concentration after area variation in the magnetic image suggests that ACB was able to assess the coat dissolution and tablet disintegration in vivo.

Magnetic images and pharmacokinetic analysis after administration of omeprazole for the same volunteer (3) are shown in Figure 5. As expected, initially (Figure 5A, t = 0 min), the coating of the tablet was intact. However, even at t = 30 min after administration (Figure 5D), the coating remained intact with omeprazole administration. Neither the magnetic area nor the metronidazole concentration showed an increase during the measurement time, suggesting that the coated magnetic tablet remained intact for this volunteer. As one may observe from Table 3, volunteer 3’s data shows that no metronidazole was detected in the plasma during the 300 min measurements after the omeprazole administration.

Our pharmacomagnetography data indicate the influence on metronidazole’s bioavailability of omeprazole concomitant administration, provoking an irregular drug absorption. We observed three different situations regarding the effects of omeprazole administration through our data of the omeprazole administration phase.

Likely due to the intra-individual biological variation, the omeprazole did not increase the pH elevation, which is endorsed by the pharmacokinetic parameters being very similar to the Control Phase of volunteers 1, 7, and 12. Other volunteers, such as 4, 6, 9, and 10, presented a delayed-release and lower bioavailability, represented by Tlag and the AUC parameters, respectively. However, due to the delayed-release (more than 30 min of monitoring by MC-ACB), the $T_{50}$ was not quantified (not available).

On the other hand, the pharmacokinetic parameters of volunteers 2, 3, 5, 8, and 11 imply that the magnetic-coated tablets were entirely emptied, with no release and consequently no absorption (even in the small intestine). The pharmacomagnetography analysis employing the MC-ACB system presented several advantages compared to other biomagnetic techniques used for this purpose. The main techniques use a superconducting quantum interference device (SQUID), which requires a magnetically shielded room to perform measurements. This restriction makes the technique more expensive and less portable and restricts its biomedical applications. In this context, other methods have been applied to evaluate, in vivo, the behavior of an oral solid dosage form in the gastrointestinal tract. These visualization methods include γ-scintigraphy, X-ray imaging, and magnetic resonance imaging. However, as with the SQUID, the drawbacks of these techniques include large radiation exposure, demand for a shielded room, high cost associated with maintenance, and specifically for MRI, and the protocol of real-time imaging acquisition is affected by the long acquisition time (at least 20 s), in which any motion on the gastric projection must be avoided to obtain a high-resolution image.

The MC-ACB system does not require electromagnetic shields, and for this reason, it is portable, versatile, and less expensive to implement [19,50]. However, in our study, the ACB system had a few drawbacks, such as the time limit of MC-ACB measurements (the volunteer must remain in an upright position for a long time after an overnight fast), spatial resolution, and the distance dependence of the signal, which restricts its application to volunteers with a low body mass index. Nevertheless, ACB is feasible for laboratory conditions, since it is portable and easily transportable between facilities to perform measurements [22].

## 4. Conclusions

The ACB system and the standard pharmacokinetics method allowed tablet movement characterization and delineated the release sites in the gastrointestinal tract. The ACB was suitable for determining the onset of the tablet’s disintegration process. In the case of elevated gastric pH associated with the use of omeprazole, the evidence suggests that the bioavailability of metronidazole is influenced by its release in the wrong place when the tablet coating is ph-dependent.

Our study reinforced the advantages of pharmacomagnetography as an attractive tool to perform a multi-instrumental correlation analysis of the disintegration process and the bioavailability of a drug. The association with pharmacokinetic parameters enables the tablet’s location at which the drug is released to be determined.

Based on our results, concomitant PPI administration with other drugs requires extensive investigation, which the ACB system may carry out through in vitro and in vivo analysis. In this way, our study showed the influence of omeprazole on tablets coated with E100.

## Figures and Tables

**Figure 1 pharmaceutics-13-01274-f001:**
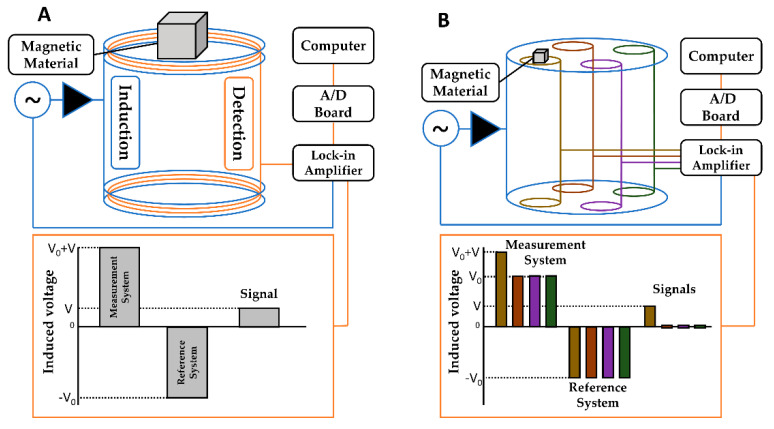
Functional diagram of the ACB setup and its principle of operation. (**A**) Schematic representation of the SC-ACB and (**B**) Schematic representation of MC-ACB with multiple detection coils.

**Figure 2 pharmaceutics-13-01274-f002:**
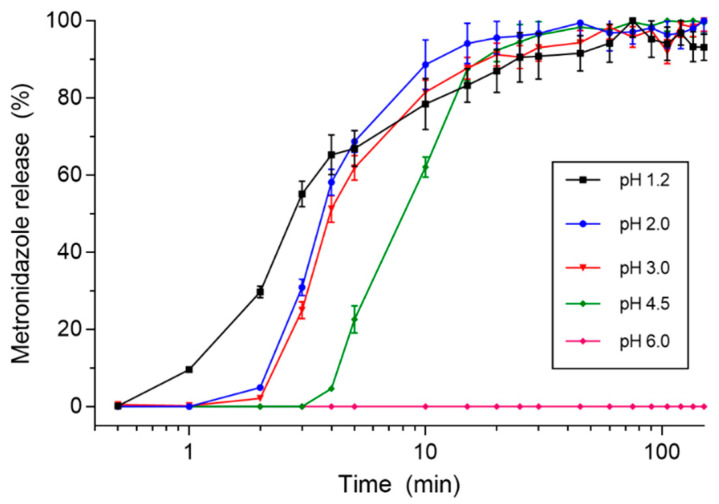
Metronidazole’s in vitro release profile versus time for distinct pH mediums. (*n* = 6).

**Figure 3 pharmaceutics-13-01274-f003:**
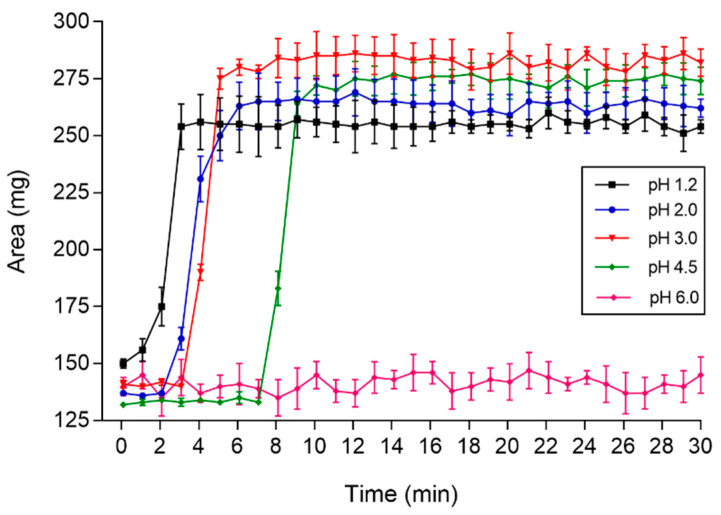
Image area of the coated magnetic tablet versus time for distinct pH mediums employing the MC-ACB (*n* = 6).

**Figure 4 pharmaceutics-13-01274-f004:**
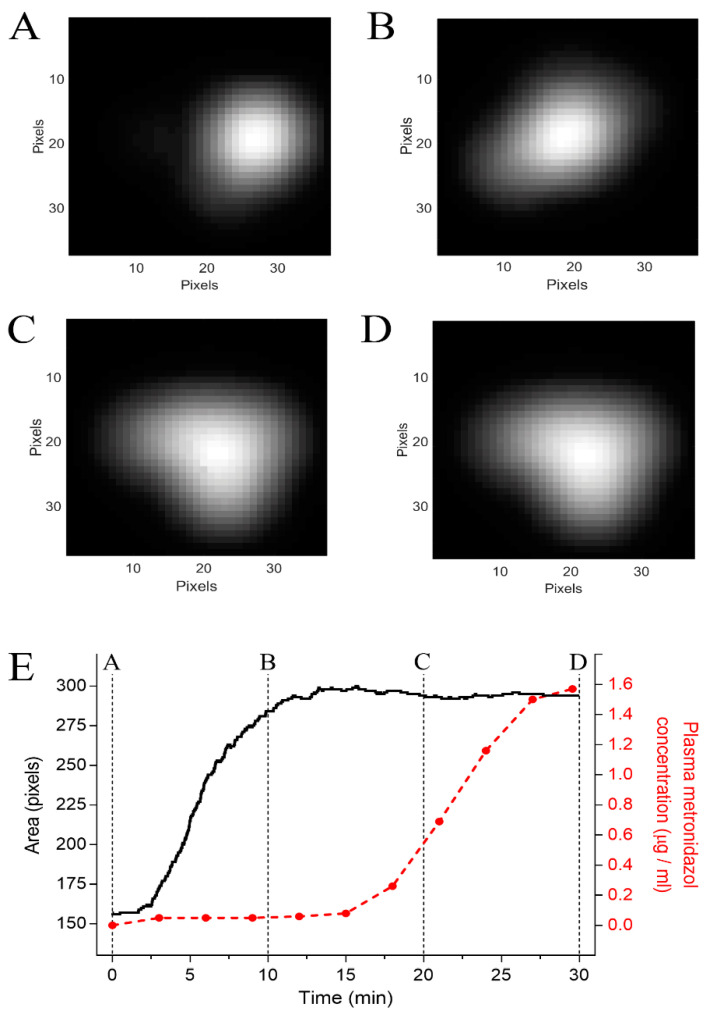
Pharmacomagnetography monitoring of the coated magnetic tablet after administration of the placebo. (**A**) Magnetic images at t = 0 min, (**B**) t= 10 min, (**C**) t= 20 min, and (**D**) t = 30 min after coated magnetic tablet administration. (**E**) Quantification of the magnetic area and metronidazole concentration, the dotted lines indicate the time points in which A, B, C, and D were captured.

**Figure 5 pharmaceutics-13-01274-f005:**
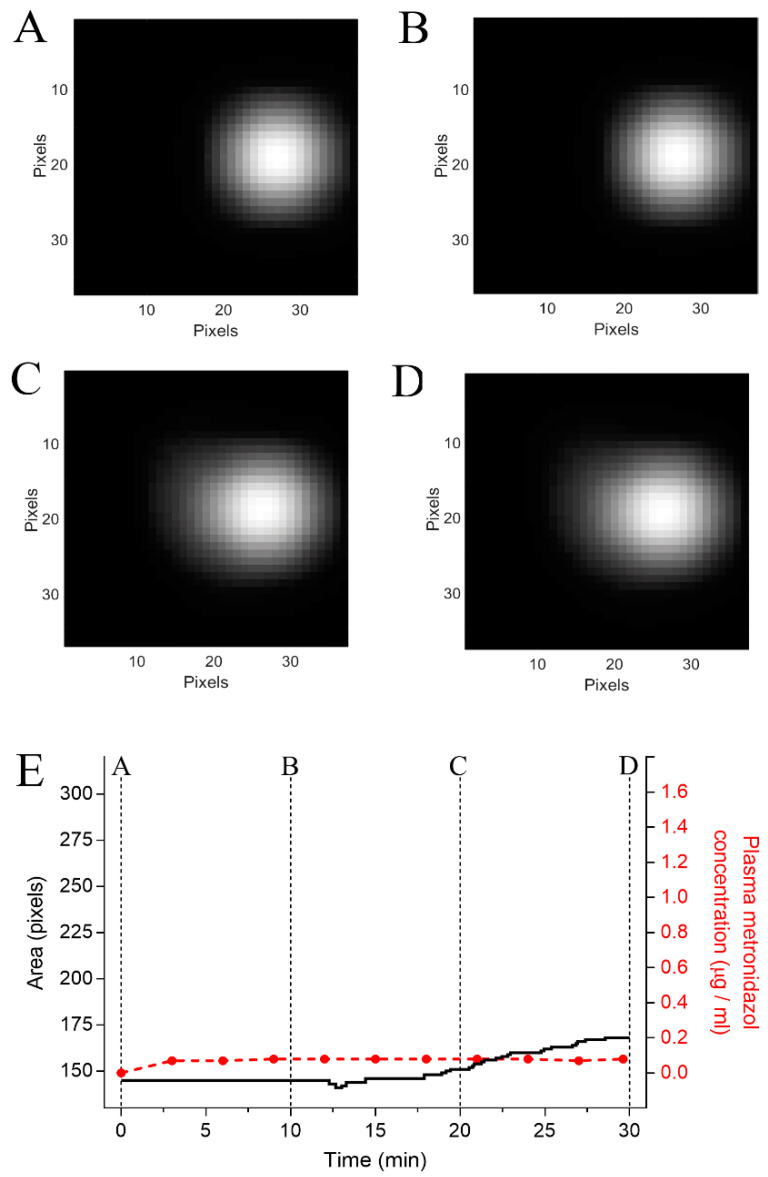
Pharmacomagnetography monitoring of the coated magnetic tablet after administration of omeprazole. (**A**) Magnetic images at t = 0 min, (**B**) t = 10 min, (**C**) t = 20 min, and (**D**) t = 30 min after coated magnetic tablet administration. (**E**) Quantification of the magnetic area and metronidazole concentration, the dotted lines indicate the time points in which A, B, C, and D were captured.

**Table 1 pharmaceutics-13-01274-t001:** Lag time ($T_{lag}$) and dissolution efficiency (DE) for coated magnetic tablets in different pH dissolution mediums (*n* = 6).

	pH 1.2	pH 2.0	pH 3.0	pH 4.5
$T_{lag}$ (min)	0.58 ± 0.20	1.16 ± 0.40	1.33 ± 0.52 ^a^	3.16 ± 0.40 ^a,b,c^
DE (%)	90.93 ± 8.11	94.21 ± 4.55	92.21 ± 2.51	92.50 ± 5.08

^a^*p* < 0.05 in comparison to pH 1.2.; ^b^
*p* < 0.05 in comparison to pH 2.0.; ^c^
*p* < 0.05 in comparison to pH 3.0.

**Table 2 pharmaceutics-13-01274-t002:** The onset of the disintegration process ($t_{50}$) for distinct pH mediums (*n* = 6).

	pH 1.2	pH 2.0	pH 3.0	pH 4.5
$t_{50}$ (min)	2.19 ± 0.10	3.03 ± 0.18 ^a^	4.01 ± 0.15 ^a,b^	9.54 ± 1.42 ^a,b,c^

^a^ *p* < 0.05 in comparison to pH 1.2.; ^b^ *p* < 0.05 in comparison to pH 2.0.; ^c^ *p* < 0.05 in comparison to pH 3.0.

**Table 3 pharmaceutics-13-01274-t003:** Comparison of pharmacokinetic parameters before (control) and after omeprazole administration to healthy volunteers. Data are expressed as the mean ± SD and median. NA= not available * *p* < 0.05 vs. control; ** *p*< 0.001.

			Control				Omeprazole							
Subject	T_lag_ (min)	T_max_ (min)	C_max_ (µg/m)	AUC_0–60_ (µg min/mL)	AUC_0–300_ (µg min/mL)	T_50_ (min)	GRT (min)	T_lag_ (min)	T_max_ (min)	C_max_ (µg/mL)	AUC_0–60_ (µg min/mL)	AUC_0–300_ (µg min/mL)	T_50_ (min)	GRT (min)
1	18.00	90.00	1.99	38.76	500.91	7.00	35.00	12.00	120.00	7.64	32.99	527.69	10.00	45.00
2	15.00	150.00	1.69	37.28	420.08	10.00	40.00	NA	0	0	0	0	NA	35.00
3	18.00	120.00	2.49	59.759	556.34	7.00	45.00	NA	0	0	0	0	NA	40.00
4	18.00	150.00	2.42	18.90	534.65	8.00	40.00	45.00	180.00	1.83	18.86	512.21	NA	35.00
5	30.00	90.00	2.57	55.35	531.90	23.00	45.00	NA	0	0	0	0	NA	75.00
6	15.00	90.00	2.80	83.36	506.36	12.00	105.00	150.00	240.00	2.68	5.31	305.61	NA	120.00
7	15.00	120.00	3.10	62.49	718.44	10.00	120.00	21.00	120.00	5.42	54.39	773.49	10.00	120.00
8	24.00	120.00	1.55	27.72	345.15	18.00	60.00	NA	0	0	0	0	NA	40.00
9	24.00	90.00	1.91	26.52	545.47	15.00	40.00	45.00	150.00	3.09	21.09	507.09	NA	35.00
10	18.00	120.00	2.20	40.62	484.92	12.00	45.00	45.00	90.00	2.46	14.09	452.09	NA	35.00
11	18.00	120.00	1.80	27.86	353.96	8.00	35.00	NA	0	0	0	0	NA	50.00
12	18.00	120.00	1.52	36.68	363.83	10.00	45.00	15.00	90.00	9.55	35.58	441.03	10.00	35.00
Mean	19.25	115.00	2.17	42.76	488.50	11.67	54.58	27.75	82.5	1.333	15.19	293.3	-	55.42
SD	4.51	21.53	0.51	18.55	106.20	4.85	28.00	42.79	83.03	1.25	18.02	279	-	32.22
Median	18.00	120.00	2.09	38.02	503.60	10.00	45.00	13.5	90.00	1.86	9.7	373.3	-	40.00
IQ	15.75-22.5	90-120	1.718-2.55	27.76–57.03	377.9–543		40–56.25	0–45	0–142.5	0–2.173	0–30.02 **	0–510.9 *		35–98.75

## Data Availability

Almost all data are presented within the manuscript (figures and tables). The raw data presented in this study are available on request from the corresponding author.

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
