# Peer review of "The Influence of Omeprazole on the Dissolution Processes of pH-Dependent Magnetic Tablets Assessed by Pharmacomagnetography"

_pharmaceutics, 2021, doi:10.3390/pharmaceutics13081274_

Round 1
Reviewer 1 Report
Dear Authors,
Many thanks for providing the opportunity to review your manuscript entitled "Influence of omeprazole on dissolution processes of pH-dependent magnetic tablets assessed by pharmacomagnetography". Firstly, it is important to stress that this field is not my area of expertise, but I have spent some time understanding the field before giving my full review below. Overall, the publication provides an interesting read and the methodology described could have many useful applications in the design and delivery of pH-dependent oral formulations. In particular, it would be useful to see some future work that looks at drugs that have absorption sites further along the intestines to see if the methodology described within the manuscript would be suitable for identifying modified-release/delayed-release and absorption at these later sites also. Please find below my comments which may help to improve the overall quality of the publication.
- On giving the manuscript a complete read, it appears as though there have been multiple authors involved in the writing of the publication - evidenced by the various writing styles and varying English terms, grammar and sentence structures used throughout. Please ensure that one author has thoroughly read the manuscript and made the necessary changes to ensure that the publication flows and reads as a single piece, rather than multiple pieces as it does currently.
- Eudragit® E-100 is mentioned multiple times throughout the manuscript. It is however referred to in multiple ways: Eudragit E-100, Eudragit® E-100, Eudragit® E-100 or E-100/E100. The first mention of Eudragit® E-100 is on line 55, with an abbreviation (E100) provided on line 56 after the second mention of Eudragit® E-100. Please provide consistent terms and/or single abbreviation for Eudragit® E-100 throughout the manuscript, including the abstract.
- Solid pharmaceutical forms is first abbreviated in line 39 to SPFs. There are times in the manuscript that SPFs are referred to in full. Either abbreviate or always provide the full term.
- When referring to in vitro and in vivo, the terms are sometimes in italics (i.e., in vitro and in vivo) and sometimes in normal text. Please keep this consistent.
- In section 2.1, you provide abbreviations for metronidazole, isopropyl alcohol, acetonitrile and polyethylene glycol. These are never used in the remainder of the manuscript. Please remove, or use.
- In the final sentence of section 2.2, you refer to a coating procedure that was first identified in the manufacturer's technical information. The reference is a book, which is 20 years old and is difficult to get a hold of. A sentence or couple of sentences explaining the coating methodology to the reader based on this reference would go a long way to improving the reader's understanding.
- In section 2.3, you discuss the methodology of the two ACB setups. The reader would be interested in and would benefit from seeing a diagrammatic representation explaining each of these setups (SC-ACB and MC-ACB).
- Line 121, after defining gastric retention time (GRT), the description is missing a closing bracket after elimination in the stomach.
- Lines 124 and 125, the reference style appears different - the brackets used are rounded as opposed to square brackets seen throughout the manuscript.
- In section 2.4.1, you have written "An exact sample volume (1 mL) was withdrawn and replaced with a fresh dissolution medium at a predefined sample time interval". I would be very interested to know what this predefined sample time interval was, and how it was calculated. Please add a sentence(s) to explain as this is not obvious.
- Line 140, the sentence ends in a comma, please replace with a full stop.
- Line 143, has "100%" been written twice in error?
- Line 152, it states "signals were acquired for 18 min". Why was 18 minutes chosen? How was this calculation determined. Please add a sentence(s) explaining your methodology and reasoning.
- Section 2.5. describes your in vivo methodology. In your limitations, your write that volunteers were required to stand during MC-ACB. Please include this in Section 2.5. as this is not clear to the reader (I assumed participants were lying down).
- Figure 1 & Figure 2, please keep the key (colour of lines and shape) consistent with pH values, as they currently differ between the two figures. Also, try to match the size (dimensions) of the two figures as Figure 1 is currently smaller than Figure 2.
- All Figures and Tables, please include sample numbers (n=x) in captions to remind the reader of the number of samples (participants) involved in your study and data analysis.
- Line 286, you discuss the use of Biopharmaceutical Classification System (BCS), but the relevance of this sentence here is not clear, nor the meaning/explanation you are trying to provide. If you are keeping this sentence here, please provide a further explanation and reference. Alternatively, this sentence appears more relevant on line 341 (beneath Table 3). In either case, explain and reference accordingly.
- Lines 325-327, you write "If no magnetic data were collected, one could conclude that the coating was dissolved in the stomach. However, the pharmacomagnetography analysis may state that the coating was dissolved." Why are these two sentences linked with a however? Are you not making the point that if there is no magnetic data in the small intestines, then the coating has dissolved in the stomach, which could be detected with pharmacomagnetography?
- Lines 327-329, you write "The metronidazole was probably absorbed in the small intestine, endorsed by a previous study that showed the upper part of the small intestine is the leading site of release of metronidazole". The leading site of release does not necessarily correlate with absorption site of metronidazole, since your study is looking at the release of metronidazole from E100 coated tablets, which as you have shown is pH dependent. Therefore can these be the same?
- Line 334, you state "intragastric pH was likely above 5, which is the upper limit for E100 solubility". Is it possible to make this statement when the highest limit of pH tested for is 4.5?
- Lines 348, you talk about "impact on drug bioavailability". Is this not obvious? Low permeability will reduce drug bioavailability even without omeprazole present. Therefore what are you hoping to achieve with omeprazole co-administration for BCS class II and IV drugs? An explanation here just before presenting the Sonidegib case would be of benefit.
- Line 366/367, you refer to "taste masking". How often are pH-dependent coatings used in taste masking paediatric formulations? If there are some, please provide examples that would need to be avoided when omeprazole is administered concomitantly as this would be of significant interest at the end of Line 369.
- Line 370, you refer to "inadequate disorders". What do you mean here? This term is often used to describe mental health disorders, not those of the gastro-intestinal tract that you seem to be referring to.
- Line 372, you refer to "antiviral and antifungals". What are these drugs more susceptible to adverse outcomes than other drug classes when used with PPIs? Omeprazole is known to decrease the effects of antivirals, but there is no documented interactions with antifungals (of great significance).
- Lines 380-381, "coating was first dissolved after the metronidazole was detected in the plasma". Figure 3C would indicate that plasma metronidazole concentration increases after E100 coating dissolves? Wouldn't the coating have to dissolve first anyway before drug release would be detected?
- Figure 4 and explanation above is for volunteer 3. Where the results also true for volunteers 2,5,8 and 11? It would be interesting to see the data for all 5 participants (does not have to be included, but should be discussed at least). Also, the results suggest that the absorption site (upper small intestine as you refer to earlier in the manuscript) was missed altogether due to delayed release from the coating?
Please feel free to address these comments. Would be very interested to see the final version, as overall, a very interesting read and novel.
Reviewer 2 Report
In this work, the author investigated the biomagnetic technique for understanding the performances of solid pharmaceutical formulations. The aim of this study was to use pharmacomagnetography to evaluate the effect of gastric pH changes caused by treatment with omeprazole in the process of metronidazole releasing administered orally in tablets coated with Eudragit E-100. It is so predictable, and usually, a class of methacrylic polymers such as Eudragit E is used to mask the bitter taste of drugs. Therefore, the content and rationale of the manuscript have poor scientific discussions in terms of pharmaceutics. The quality of the submitted work seems to be no in-depth understanding for satisfying the criteria of pharmaceutical drug development. Therefore, I would reject and encourage the resubmission of the manuscript.
- The authors should present a detailed description of other non-invasive analysis methods that are often applied to improve the quality and the performances of drug formulations through NMR or NIR imaging, but more specific examples and explanations are required. A thorough comparison will doubtless improve the discussion extent of the current research article related to non-invasive insight into the release mechanisms of solid pharmaceutical formulations (https://doi.org/10.1016/j.ejpb.2016.02.001, https://doi.org/10.1002/jrs.4896)
- In order to further support the potential of this formulation as pharmaceutical drug development, it is necessary to explain the standardized experimental method that should follow the US pharmacopeia with the physicochemical properties of the pharmaceutical product.
- The stability conditions related to the active pharmaceutical ingredient need to be monitored for three batches manufactured under long-term storage conditions and accelerated test conditions, although 100% of the ICH guidelines cannot be followed by these kinds of lab-scale experiments. In this study, the stability results confirmed that the significant stability of the active pharmaceutical ingredient in the form of stress conditions, but it is not sufficient as a stability result that can be supported in terms of quality management.
- The current direction of pharmaceutical science research is to examine the attractive tool with the pharmacomagnetography for identifying the performances of drug formulation, and finally to evaluate the bioavailability in a clinical trial. The present authors have achieved very meaningful results. However, the extent of discussion and in-depth mechanism are very poor.
Reviewer 3 Report
The authors propose in this paper to assess the suitability of a new biomagnetic technique, Alternate Current Biosusceptometry (ACB) as an alternative method for the in vivo evaluation of some solid pharmaceutical forms behavior, along with different segments of the gastrointestinal tract.
In a first step, the pH dependence of the dissolution of tablets containing metronidazole, and magnetic particles coated with Eutragit -E 100 (a polymer with a pH-dependent solubility) was assessed by using both a standard method and ACB and a good correlation between the two methods was obtained.
For the in vivo study, the authors used the ACB technique combined with standard pharmacokinetic analysis (in an approach called Pharmacomagnetography) for observing the effects of a proton pump inhibitor (omeprazole) on the metronidazole release from orally administered tablets in 12 volunteers.
The obtained results indicate an increased dissolution time of Eudragit® E-100 coated tablets with increasing pH. The in vivo results suggest that omeprazole treatment interferes in the release process of Metronidazole reducing its bioavailability.
The paper is well written the experimental design and the characterization techniques were correctly chosen, the results are interesting and promising and, therefore, I endorse its publication.
There are some minor, more or less formal observations the authors might address before publication:
- there are small, minor English errors;
- the paper is based on some previous works published by the authors; for the sake of readability, I suggest the authors provide more minimum details about their ACB setup (spatial resolution, sensitivity, magnetic data analysis) so that a reader understands without the need to check their previous works.
Reviewer 4 Report
In this manuscript, the authors developed metronidazole-magnetic tablets coated with Eudragit E-100, and used pharmacomagnetography to prove that gastric pH changes can influence the dissolution and bioavailability of metronidazole administered orally in magnetic tablets. Especially, the authors observed the drug dissolution process by pharmacomagnetography. It is a topic of interest to the scientists in the related areas but the manuscript needs improvement before acceptance for publication.
The detailed comments are listed as follows:
- In figure 1, all coated magnetic tablets were disintegrated entirely during the measurement time, regardless of the pH value, but the coated magnetic tablets remained intact during 30 min in gastric fluid after omeprazole administration in in vivo studies protocol. Therefore, the authors should supplement an in vitro study to prove the intactness of magnetic tablets (pH >5).
- In Page 4, authors analyzed the samples of dissolution studies by UV-spectrophotometer at 277 nm (pH 1.2 and pH 2.0) or 318 nm (pH 3.0 and pH 4.5). Please explain the reason of using different wavelength in different pH values.
- In Page 5, at night before the experiment, the volunteers received either 40 mg of placebo or 40 mg of immediate-release omeprazole tablet. But others administrated omeprazole just 20 mg in following references (page 9, reference 35). Do the authors have any specific reason using 40 mg omeprazole?
- In figure 2, the maximum of Image area was inconsistent at different pH, but all coated magnetic tablets were disintegrated entirely during the measurement time, regardless of the pH value. So, please explain this abnormal phenomenon, some experiments may be supplemented if necessary.
- In figure 3 and figure 4, please add more images at different time after coated magnetic tablet administration to show the process of drug
- In table 3, the mean Cmax and AUC0-300 of omeprazole group was higher than control group. This is contradictory with authors’ description in page 9, line 331-334. Please explain and cite some references.
- In page 9, the conclusion between line 325 and line 326 was contradict, please check and explain.
8. Please unify the format the title of table, including font size, space, etc.
Round 2
Reviewer 1 Report
Dear Authors,
Much has been changed since the original manuscript was submitted. Despite the amendments, there are some new outstanding issues that require addressing, as described below:
1. Lines 123-134, you describe the calculation of tablet mass and friability in detail, however the results are not mentioned anywhere within the manuscript. Please include and discuss these results within the manuscript.
2. In section 3.1, you refer to Figure 1, which has now been renamed to Figure 2.
3. Figure 2 and Figure 3 refer to n=6 (as stated in your methods), however, Table 1 and Table 2 refer to n=3. Why has the number of samples analysed reduced? Have you been selective about the data chosen for analysis in the Tables? Please comment and justify why this has been changed.
Please address these at a minimum before resubmitting.
Reviewer 4 Report
After modification, the authors supplemented some experiments to support their conclusion both in vivo and in vitro according to our comments, which makes the research more reasonable. Therefore, we recommend to be published in pharmaceutics.
Round 3
Reviewer 1 Report
Thank you for your corrections based upon the comments provided previously. I am happy with this version of the manuscript.
